# Sparse Random Features Algorithm as Coordinate Descent in Hilbert Space

**Ian E.H. Yen** [1]   **Ting-Wei Lin** [2]   **Shou-De Lin** [2]   **Pradeep Ravikumar** [1]   **Inderjit S. Dhillon** [1]

Department of Computer Science

1: University of Texas at Austin,    2: National Taiwan University

1: {ianyen,pradeepr,inderjit}@cs.utexas.edu,

2: {b97083,sdlin}@csie.ntu.edu.tw

## Abstract

In this paper, we propose a Sparse Random Features algorithm, which learns a sparse non-linear predictor by minimizing an $\ell_1$-regularized objective function over the Hilbert Space induced from a kernel function. By interpreting the algorithm as Randomized Coordinate Descent in an infinite-dimensional space, we show the proposed approach converges to a solution within $\epsilon$-precision of that using an exact kernel method, by drawing $O(1/\epsilon)$ random features, in contrast to the $O(1/\epsilon^2)$ convergence achieved by current Monte-Carlo analyses of Random Features. In our experiments, the Sparse Random Feature algorithm obtains a sparse solution that requires less memory and prediction time, while maintaining comparable performance on regression and classification tasks. Moreover, as an approximate solver for the infinite-dimensional $\ell_1$-regularized problem, the randomized approach also enjoys better convergence guarantees than a Boosting approach in the setting where the greedy Boosting step cannot be performed exactly.

## 1   Introduction

Kernel methods have become standard for building non-linear models from simple feature representations, and have proven successful in problems ranging across classification, regression, structured prediction and feature extraction [16, 20]. A caveat however is that they are not scalable as the number of training samples increases. In particular, the size of the models produced by kernel methods scale linearly with the number of training samples, even for sparse kernel methods like support vector machines [17]. This makes the corresponding training and prediction computationally prohibitive for large-scale problems.

A line of research has thus been devoted to kernel approximation methods that aim to preserve predictive performance, while maintaining computational tractability. Among these, Random Features has attracted considerable recent interest due to its simplicity and efficiency [2, 3, 4, 5, 10, 6]. Since first proposed in [2], and extended by several works [3, 4, 5, 10], the Random Features approach is a sampling based approximation to the kernel function, where by drawing $D$ features from the distribution induced from the kernel function, one can guarantee uniform convergence of approximation error to the order of $O(1/\sqrt{D})$. On the flip side, such a rate of convergence suggests that in order to achieve high precision, one might need a large number of random features, which might lead to model sizes even larger than that of the vanilla kernel method.

One approach to remedy this problem would be to employ feature selection techniques to prevent the model size from growing linearly with $D$. A simple way to do so would be by adding $\ell_1$-regularization to the objective function, so that one can simultaneously increase the number of random features $D$, while selecting a compact subset of them with non-zero weight. However, the resulting algorithm cannot be justified by existing analyses of Random Features, since the Representer theorem does not hold for the $\ell_1$-regularized problem [15, 16]. In other words, since the prediction

cannot be expressed as a linear combination of kernel evaluations, a small error in approximating the kernel function cannot correspondingly guarantee a small prediction error.

In this paper, we propose a new interpretation of Random Features that justifies its usage with $\ell_1$-regularization — yielding the Sparse Random Features algorithm. In particular, we show that the Sparse Random Feature algorithm can be seen as Randomized Coordinate Descent (RCD) in the Hilbert Space induced from the kernel, and by taking $D$ steps of coordinate descent, one can achieve a solution comparable to exact kernel methods within $O(1/D)$ precision in terms of the objective function. Note that the surprising facet of this analysis is that in the finite-dimensional case, the iteration complexity of RCD increases with number of dimensions [18], which would trivially yield a bound going to infinity for our infinite-dimensional problem. In our experiments, the Sparse Random Features algorithm obtains a sparse solution that requires less memory and prediction time, while maintaining comparable performance on regression and classification tasks with various kernels. Note that our technique is complementary to that proposed in [10], which aims to reduce the cost of evaluating and storing basis functions, while our goal is to reduce the number of basis functions in a model.

Another interesting aspect of our algorithm is that our infinite-dimensional $\ell_1$-regularized objective is also considered in the literature of Boosting [7, 8], which can be interpreted as greedy coordinate descent in the infinite-dimensional space. As an approximate solver for the $\ell_1$-regularized problem, we compare our randomized approach to the boosting approach in theory and also in experiments. As we show, for basis functions that do not allow exact greedy search, a randomized approach enjoys better guarantees.

## 2   Problem Setup

We are interested in estimating a prediction function $f \colon \mathcal{X} \to \mathcal{Y}$ from training data set $\mathcal{D} = \{(\boldsymbol{x}_n, y_n)\}_{n=1}^N$, $(\boldsymbol{x}_n, y_n) \in \mathcal{X} \times \mathcal{Y}$ by solving an optimization problem over some Reproducing Kernel Hilbert Space (RKHS) $\mathcal{H}$:

$$f^* = \underset{f \in \mathcal{H}}{arg\,min} \quad \frac{\lambda}{2} \|f\|_{\mathcal{H}}^2 + \frac{1}{N} \sum_{n=1}^N L(f(\boldsymbol{x}_n), y_n), \tag{1}$$

where $L(z, y)$ is a convex loss function with Lipschitz-continuous derivative satisfying $|L'(z_1, y) - L'(z_2, y)| \leq \beta |z_1 - z_2|$, which includes several standard loss functions such as the *square-loss* $L(z, y) = \frac{1}{2}(z - y)^2$, *square-hinge loss* $L(z, y) = \max(1 - zy, 0)^2$ and *logistic loss* $L(z, y) = \log(1 + \exp(-yz))$.

### 2.1   Kernel and Feature Map

There are two ways in practice to specify the space $\mathcal{H}$. One is via specifying a positive-definite kernel $k(\boldsymbol{x}, \boldsymbol{y})$ that encodes similarity between instances, and where $\mathcal{H}$ can be expressed as the completion of the space spanned by $\{k(\boldsymbol{x}, \cdot)\}_{\boldsymbol{x} \in \mathcal{X}}$, that is,

$$\mathcal{H} = \left\{ f(\cdot) = \sum_{i=1}^K \alpha_i k(\boldsymbol{x}_i, \cdot) \mid \alpha_i \in \mathbb{R}, \boldsymbol{x}_i \in \mathcal{X} \right\}.$$

The other way is to find an explicit feature map $\{\bar{\phi}_h(\boldsymbol{x})\}_{h \in H}$, where each $h \in H$ defines a basis function $\bar{\phi}_h(\boldsymbol{x}) : \mathcal{X} \to \mathbb{R}$. The RKHS $\mathcal{H}$ can then be defined as

$$\mathcal{H} = \left\{ f(\cdot) = \int_{h \in H} w(h) \bar{\phi}_h(\cdot) dh = \langle \boldsymbol{w}, \bar{\boldsymbol{\phi}}(\cdot) \rangle_{\mathcal{H}} \mid \|f\|_{\mathcal{H}}^2 < \infty \right\}, \tag{2}$$

where $w(h)$ is a weight distribution over the basis $\{\phi_h(\boldsymbol{x})\}_{h \in \mathcal{H}}$. By Mercer's theorem [1], every positive-definite kernel $k(\boldsymbol{x}, \boldsymbol{y})$ has a decomposition s.t.

$$k(\boldsymbol{x}, \boldsymbol{y}) = \int_{h \in H} p(h) \phi_h(\boldsymbol{x}) \phi_h(\boldsymbol{y}) dh = \langle \bar{\boldsymbol{\phi}}(\boldsymbol{x}), \bar{\boldsymbol{\phi}}(\boldsymbol{y}) \rangle_{\mathcal{H}}, \tag{3}$$

where $p(h) \geq 0$ and $\bar{\phi}_h(.) = \sqrt{p(h)} \phi_h(.)$, denoted as $\bar{\boldsymbol{\phi}} = \sqrt{\boldsymbol{p}} \circ \boldsymbol{\phi}$. However, the decomposition is not unique. One can derive multiple decompositions from the same kernel $k(\boldsymbol{x}, \boldsymbol{y})$ based on

different sets of basis functions $\{\phi_h(\boldsymbol{x})\}_{h\in H}$. For example, in [2], the Laplacian kernel $k(\boldsymbol{x},\boldsymbol{y}) = \exp(-\gamma\|\boldsymbol{x}-\boldsymbol{y}\|_1)$ can be decomposed through both the Fourier basis and the Random Binning basis, while in [7], the Laplacian kernel can be obtained from the integrating of an infinite number of decision trees.

On the other hand, multiple kernels can be derived from the same set of basis functions via different distribution $p(h)$. For example, in [2, 3], a general decomposition method using Fourier basis functions $\left\{\phi_\omega(\boldsymbol{x}) = \cos(\omega^T\boldsymbol{x})\right\}_{\omega\in\mathbb{R}^d}$ was proposed to find feature map for any *shift-invariant* kernel of the form $k(\boldsymbol{x}-\boldsymbol{y})$, where the feature maps (3) of different kernels $k(\Delta)$ differ only in the distribution $p(\omega)$ obtained from the Fourier transform of $k(\Delta)$. Similarly, [5] proposed decomposition based on polynomial basis for any dot-product kernel of the form $k(\langle\boldsymbol{x},\boldsymbol{y}\rangle)$.

### 2.2 Random Features as Monte-Carlo Approximation

The standard kernel method, often referred to as the "kernel trick," solves problem (1) through the Representer Theorem [15, 16], which states that the optimal decision function $f^* \in \mathcal{H}$ lies in the span of training samples $\mathcal{H}_\mathcal{D} = \left\{f(\cdot) = \sum_{n=1}^N \alpha_n k(\boldsymbol{x}_n,\cdot) \mid \alpha_n \in \mathbb{R}, (\boldsymbol{x}_n,y_n) \in \mathcal{D}\right\}$, which reduces the infinite-dimensional problem (1) to a finite-dimensional problem with $N$ variables $\{\alpha_n\}_{n=1}^N$. However, it is known that even for loss functions with dual-sparsity (e.g. hinge-loss), the number of non-zero $\alpha_n$ increases linearly with data size [17].

Random Features has been proposed as a kernel approximation method [2, 3, 10, 5], where a Monte-Carlo approximation

$$k(\boldsymbol{x}_i,\boldsymbol{x}_j) = E_{p(h)}[\phi_h(\boldsymbol{x}_i)\phi_h(\boldsymbol{x}_j)] \approx \frac{1}{D}\sum_{k=1}^D \phi_{h_k}(\boldsymbol{x}_i)\phi_{h_k}(\boldsymbol{x}_j) = \boldsymbol{z}(\boldsymbol{x}_i)^T\boldsymbol{z}(\boldsymbol{x}_j) \qquad (4)$$

is used to approximate (3), so that the solution to (1) can be obtained by

$$\boldsymbol{w}_{RF} = \underset{\boldsymbol{w}\in\mathbb{R}^D}{argmin}\quad \frac{\lambda}{2}\|\boldsymbol{w}\|^2 + \frac{1}{N}\sum_{n=1}^N L(\boldsymbol{w}^T\boldsymbol{z}(\boldsymbol{x}_n),y_n). \qquad (5)$$

The corresponding approximation error

$$\left|\boldsymbol{w}_{RF}^T\boldsymbol{z}(\boldsymbol{x}) - f^*(\boldsymbol{x})\right| = \left|\sum_{n=1}^N \alpha_n^{RF}\boldsymbol{z}(\boldsymbol{x}_n)^T\boldsymbol{z}(\boldsymbol{x}) - \sum_{n=1}^N \alpha_n^* k(\boldsymbol{x}_n,\boldsymbol{x})\right|, \qquad (6)$$

as proved in [2,Appendix B], can be bounded by $\epsilon$ given $D = \Omega(1/\epsilon^2)$ number of random features, which is a direct consequence of the uniform convergence of the sampling approximation (4). Unfortunately, the rate of convergence suggests that to achieve small approximation error $\epsilon$, one needs significant amount of random features, and since the model size of (5) grows linearly with $D$, such an algorithm might not obtain a sparser model than kernel method. On the other hand, the $\ell_1$-regularized Random-Feature algorithm we are proposing aims to minimize loss with a selected subset of random feature that neither grows linearly with $D$ nor with $N$. However, (6) does not hold for $\ell_1$-regularization, and thus one cannot transfer guarantee from kernel approximation (4) to the learned decision function.

## 3 Sparse Random Feature as Coordinate Descent

In this section, we present the Sparse Random Features algorithm and analyze its convergence by interpreting it as a fully-corrective randomized coordinate descent in a Hilbert space. Given a feature map of orthogonal basic functions $\{\bar{\phi}_h(\boldsymbol{x}) = \sqrt{p(h)}\phi_h(\boldsymbol{x})\}_{h\in H}$, the optimization program (1) can be written as the infinite-dimensional optimization problem

$$\min_{\boldsymbol{w}\in\mathcal{H}}\quad \frac{\lambda}{2}\|\boldsymbol{w}\|_2^2 + \frac{1}{N}\sum_{n=1}^N L(\langle\boldsymbol{w},\bar{\boldsymbol{\phi}}(\boldsymbol{x}_n)\rangle_\mathcal{H},y_n). \qquad (7)$$

Instead of directly minimizing (7), the *Sparse Random Features* algorithm optimizes the related $\ell_1$-regularized problem defined as

$$\min_{\bar{\boldsymbol{w}} \in \mathcal{H}} \quad F(\bar{\boldsymbol{w}}) = \lambda \|\bar{\boldsymbol{w}}\|_1 + \frac{1}{N} \sum_{n=1}^{N} L(\langle \bar{\boldsymbol{w}}, \boldsymbol{\phi}(\boldsymbol{x}_n) \rangle_{\mathcal{H}}, y_n), \tag{8}$$

where $\bar{\boldsymbol{\phi}}(\boldsymbol{x}) = \sqrt{\boldsymbol{p}} \circ \boldsymbol{\phi}(\boldsymbol{x})$ is replaced by $\boldsymbol{\phi}(\boldsymbol{x})$ and $\|\bar{\boldsymbol{w}}\|_1$ is defined as the $\ell_1$-norm in function space $\|\bar{\boldsymbol{w}}\|_1 = \int_{h \in H} |\bar{w}(h)| dh$. The whole procedure is depicted in Algorithm 1. At each iteration, we draw $R$ coordinates $h_1, h_2, ..., h_R$ from distribution $p(h)$, add them into a working set $A^t$, and minimize (8) w.r.t. the working set $A^t$ as

$$\min_{\bar{w}(h), h \in A^t} \quad \lambda \sum_{h \in A^t} |\bar{w}(h)| + \frac{1}{N} \sum_{n=1}^{N} L(\sum_{h \in A^t} \bar{w}(h) \phi_h(\boldsymbol{x}_n), y_n). \tag{9}$$

At the end of each iteration, the algorithm removes features with zero weight to maintain a compact working set.

---

**Algorithm 1** Sparse Random-Feature Algorithm

---

Initialize $\bar{\boldsymbol{w}}^0 = \boldsymbol{0}$, working set $A^{(0)} = \{\}$, and $t = 0$.
**repeat**
    1. Sample $h_1, h_2, ..., h_R$ i.i.d. from distribution $p(h)$.
    2. Add $h_1, h_2, ..., h_R$ to the set $A^{(t)}$.
    3. Obtain $\bar{\boldsymbol{w}}^{t+1}$ by solving (9).
    4. $A^{(t+1)} = A^{(t)} \setminus \{h \mid \bar{w}^{t+1}(h) = 0\}$.
    5. $t \leftarrow t + 1$.
**until** $t = T$

---

### 3.1 Convergence Analysis

In this section, we analyze the convergence behavior of Algorithm 1. The analysis comprises of two parts. First, we estimate the number of iterations Algorithm 1 takes to produce a solution $\boldsymbol{w}^t$ that is at most $\epsilon$ away from some arbitrary reference solution $\boldsymbol{w}^{ref}$ on the $\ell_1$-regularized program (8). Then, by taking $\boldsymbol{w}^{ref}$ as the optimal solution $\boldsymbol{w}^*$ of (7), we obtain an approximation guarantee for $\boldsymbol{w}^t$ with respect to $\boldsymbol{w}^*$. The proofs for most lemmas and corollaries will be in the appendix.

**Lemma 1.** *Suppose loss function $L(z, y)$ has $\beta$-Lipschitz-continuous derivative and $|\phi_h(\boldsymbol{x})| \leq B, \forall h \in \mathcal{H}, \forall \boldsymbol{x} \in \mathcal{X}$. The loss term $Loss(\bar{\boldsymbol{w}}; \boldsymbol{\phi}) = \frac{1}{N} \sum_{n=1}^{N} L(\langle \bar{\boldsymbol{w}}, \boldsymbol{\phi}(\boldsymbol{x}_n) \rangle, y_n)$ in (8) has*

$$Loss(\bar{\boldsymbol{w}} + \eta \boldsymbol{\delta}_h; \boldsymbol{\phi}) - Loss(\bar{\boldsymbol{w}}; \boldsymbol{\phi}) \leq g_h \eta + \frac{\gamma}{2} \eta^2,$$

*where $\boldsymbol{\delta}_h = \boldsymbol{\delta}(\|x - h\|)$ is a Dirac function centered at $h$, and $g_h = \nabla_{\bar{\boldsymbol{w}}} Loss(\bar{\boldsymbol{w}}; \boldsymbol{\phi})(h)$ is the Frechet derivative of the loss term evaluated at $h$, and $\gamma = \beta B^2$.*

The above lemma states smoothness of the loss term, which is essential to guarantee descent amount obtained by taking a coordinate descent step. In particular, we aim to express the expected progress made by Algorithm 1 as the *proximal-gradient* magnitude of $\bar{F}(\boldsymbol{w}) = F(\sqrt{\boldsymbol{p}} \circ \boldsymbol{w})$ defined as

$$\bar{F}(\boldsymbol{w}) = \lambda \|\sqrt{\boldsymbol{p}} \circ \boldsymbol{w}\|_1 + \frac{1}{N} \sum_{n=1}^{N} L(\langle \boldsymbol{w}, \bar{\boldsymbol{\phi}}(\boldsymbol{x}_n) \rangle, y_n). \tag{10}$$

. Let $\boldsymbol{g} = \nabla_{\bar{\boldsymbol{w}}} Loss(\bar{\boldsymbol{w}}, \boldsymbol{\phi})$, $\bar{\boldsymbol{g}} = \nabla_{\boldsymbol{w}} Loss(\boldsymbol{w}, \bar{\boldsymbol{\phi}})$ be the gradients of loss terms in (8), (10) respectively, and let $\boldsymbol{\rho} \in \partial(\lambda \|\bar{\boldsymbol{w}}\|_1)$. We have following relations between (8) and (10):

$$\bar{\boldsymbol{\rho}} := \sqrt{\boldsymbol{p}} \circ \boldsymbol{\rho} \in \partial(\lambda \|\sqrt{\boldsymbol{p}} \circ \boldsymbol{w}\|_1), \quad \bar{\boldsymbol{g}} = \sqrt{\boldsymbol{p}} \circ \boldsymbol{g}, \tag{11}$$

by simple applications of the chain rule. We then analyze the progress made by each iteration of Algorithm 1. Recalling that we used $R$ to denote the number of samples drawn in step 1 of our algorithm, we will first assume $R = 1$, and then show that same result holds also for $R > 1$.

**Theorem 1** (Descent Amount). *The expected descent of the iterates of Algorithm 1 satisfies*

$$E[F(\bar{\boldsymbol{w}}^{t+1})] - F(\bar{\boldsymbol{w}}^t) \leq -\frac{\gamma\|\bar{\boldsymbol{\eta}}^t\|^2}{2}, \tag{12}$$

*where $\bar{\boldsymbol{\eta}}$ is the proximal gradient of (10), that is,*

$$\bar{\boldsymbol{\eta}} = \underset{\boldsymbol{\eta}}{argmin} \quad \lambda\|\sqrt{\boldsymbol{p}} \circ (\boldsymbol{w}^t + \boldsymbol{\eta})\|_1 - \lambda\|\sqrt{\boldsymbol{p}} \circ \boldsymbol{w}^t\|_1 + \langle \bar{\boldsymbol{g}}, \boldsymbol{\eta} \rangle + \frac{\gamma}{2}\|\boldsymbol{\eta}\|^2 \tag{13}$$

*and $\bar{\boldsymbol{g}} = \nabla_{\boldsymbol{w}} Loss(\boldsymbol{w}^t, \bar{\boldsymbol{\phi}})$ is the derivative of loss term w.r.t. $\boldsymbol{w}$.*

*Proof.* Let $g_h = \nabla_{\bar{\boldsymbol{w}}} Loss(\bar{\boldsymbol{w}}^t, \boldsymbol{\phi})(h)$. By Corollary 1, we have

$$F(\bar{\boldsymbol{w}}^t + \eta\boldsymbol{\delta}_h) - F(\bar{\boldsymbol{w}}^t) \leq \lambda|\bar{w}^t(h) + \eta| - \lambda|\bar{w}^t(h)| + g_h\eta + \frac{\gamma}{2}\eta^2. \tag{14}$$

Minimizing RHS w.r.t. $\eta$, the minimizer $\eta_h$ should satisfy

$$g_h + \rho_h + \gamma\eta_h = 0 \tag{15}$$

for some sub-gradient $\rho_h \in \partial(\lambda|\bar{w}^t(h) + \eta_h|)$. Then by definition of sub-gradient and (15) we have

$$\lambda|\bar{w}^t(h) + \eta| - \lambda|\bar{w}^t(h)| + g_h\eta + \frac{\gamma}{2}\eta^2 \leq \rho_h\eta_h + g_h\eta_h + \frac{\gamma}{2}\eta_h^2 \tag{16}$$

$$= -\gamma\eta_h^2 + \frac{\gamma}{2}\eta_h^2 = -\frac{\gamma}{2}\eta_h^2. \tag{17}$$

Note the equality in (16) holds if $\bar{w}^t(h) = 0$ or the optimal $\eta_h = 0$, which is true for Algorithm 1. Since $\bar{\boldsymbol{w}}^{t+1}$ minimizes (9) over a block $A^t$ containing $h$, we have $F(\bar{\boldsymbol{w}}^{t+1}) \leq F(\bar{\boldsymbol{w}}^t + \eta_h\boldsymbol{\delta}_h)$. Combining (14) and (16), taking expectation over $h$ on both sides, and then we have

$$E[F(\bar{\boldsymbol{w}}^{t+1})] - F(\bar{\boldsymbol{w}}^t) \leq -\frac{\gamma}{2}E[\eta_h^2] = \|\sqrt{\boldsymbol{p}} \circ \boldsymbol{\eta}\|^2 = \|\bar{\boldsymbol{\eta}}\|^2$$

Then it remains to verify that $\bar{\boldsymbol{\eta}} = \sqrt{\boldsymbol{p}} \circ \boldsymbol{\eta}$ is the proximal gradient (13) of $\bar{F}(\boldsymbol{w}^t)$, which is true since $\bar{\boldsymbol{\eta}}$ satisfies the optimality condition of (13)

$$\bar{\boldsymbol{g}} + \bar{\boldsymbol{\rho}} + \gamma\bar{\boldsymbol{\eta}} = \sqrt{\boldsymbol{p}} \circ (\boldsymbol{g} + \boldsymbol{\rho} + \gamma\boldsymbol{\eta}) = \boldsymbol{0},$$

where first equality is from (11) and the second is from (15). □

**Theorem 2** (Convergence Rate). *Given any reference solution $\boldsymbol{w}^{ref}$, the sequence $\{\boldsymbol{w}^t\}_{t=1}^{\infty}$ satisfies*

$$E[\bar{F}(\boldsymbol{w}^t)] \leq \bar{F}(\boldsymbol{w}^{ref}) + \frac{2\gamma\|\boldsymbol{w}^{ref}\|^2}{k}, \tag{18}$$

*where $k = \max\{t - c, 0\}$ and $c = \frac{2(\bar{F}(0) - \bar{F}(\boldsymbol{w}^{ref}))}{\gamma\|\boldsymbol{w}^{ref}\|^2}$ is a constant.*

*Proof.* First, the equality actually holds in inequality (16), since for $h \notin A^{(t-1)}$, we have $w^t(h) = 0$, which implies $\lambda|w^t(h) + \eta| - \lambda|w^t(h)| = \rho\eta$, $\rho \in \partial(\lambda|w^t(h) + \eta|)$, and for $h \in A^{t-1}$ we have $\bar{\eta}_h = 0$, which gives 0 to both LHS and RHS. Therefore, we have

$$-\frac{\gamma}{2}\|\bar{\boldsymbol{\eta}}\|^2 = \underset{\boldsymbol{\eta}}{min} \quad \lambda\|\sqrt{\boldsymbol{p}} \circ (\boldsymbol{w}^t + \boldsymbol{\eta})\|_1 - \lambda\|\sqrt{\boldsymbol{p}} \circ \boldsymbol{w}^t\|_1 + \bar{\boldsymbol{g}}^T\boldsymbol{\eta} + \frac{\gamma}{2}\|\boldsymbol{\eta}\|^2. \tag{19}$$

Note the minimization in (19) is separable for different coordinates. For $h \in A^{(t-1)}$, the weight $w^t(h)$ is already optimal in the beginning of iteration $t$, so we have $\bar{\rho}_h + \bar{g}_h = 0$ for some $\bar{\rho}_h \in \partial(|\sqrt{p(h)}w(h)|)$. Therefore, $\eta_h = 0$, $h \in A^{(t-1)}$ is optimal both to $(|\sqrt{p(h)}(w(h) + \eta_h)| + \bar{g}_h\eta_h)$ and to $\frac{\gamma}{2}\eta_h^2$. Set $\eta_h = 0$ for the latter, we have

$$-\frac{\gamma}{2}\|\bar{\boldsymbol{\eta}}\|^2 = \underset{\boldsymbol{\eta}}{min} \left\{ \lambda\|\sqrt{\boldsymbol{p}} \circ (\boldsymbol{w}^t + \boldsymbol{\eta})\|_1 - \lambda\|\sqrt{\boldsymbol{p}} \circ \boldsymbol{w}^t\|_1 + \langle \bar{\boldsymbol{g}}, \boldsymbol{\eta} \rangle + \frac{\gamma}{2}\int_{h \notin A^{(t-1)}} \eta_h^2 dh \right\}$$

$$\leq \underset{\boldsymbol{\eta}}{min} \left\{ \bar{F}(\boldsymbol{w}^t + \boldsymbol{\eta}) - \bar{F}(\boldsymbol{w}^t) + \frac{\gamma}{2}\int_{h \notin A^{(t-1)}} \eta_h^2 dh \right\}$$

from convexity of $\bar{F}(\boldsymbol{w})$. Consider solution of the form $\boldsymbol{\eta} = \alpha(\boldsymbol{w}^{ref} - \boldsymbol{w}^t)$, we have

$$
\begin{aligned}
-\frac{\gamma}{2}\|\bar{\boldsymbol{\eta}}\|^2 &\leq \min_{\alpha \in [0,1]} \left\{ \bar{F}\left(\boldsymbol{w}^t + \alpha(\boldsymbol{w}^{ref} - \boldsymbol{w}^t)\right) - \bar{F}(\boldsymbol{w}^t) + \frac{\gamma \alpha^2}{2} \int_{h \notin A^{(t-1)}} (w^{ref}(h) - w^t(h))^2 dh \right\} \\
&\leq \min_{\alpha \in [0,1]} \left\{ \bar{F}(\boldsymbol{w}^t) + \alpha \left( \bar{F}(\boldsymbol{w}^{ref}) - \bar{F}(\boldsymbol{w}^t) \right) - \bar{F}(\boldsymbol{w}^t) + \frac{\gamma \alpha^2}{2} \int_{h \notin A^{(t-1)}} w^{ref}(h)^2 dh \right\} \\
&\leq \min_{\alpha \in [0,1]} \left\{ -\alpha \left( \bar{F}(\boldsymbol{w}^t) - \bar{F}(\boldsymbol{w}^{ref}) \right) + \frac{\gamma \alpha^2}{2} \|\boldsymbol{w}^{ref}\|^2 \right\},
\end{aligned}
$$

where the second inequality results from $w^t(h) = 0, h \notin A^{(t-1)}$. Minimizing last expression w.r.t. $\alpha$, we have $\alpha^* = \min \left( \frac{\bar{F}(\boldsymbol{w}^t) - \bar{F}(\boldsymbol{w}^{ref})}{\gamma \|\boldsymbol{w}^{ref}\|^2}, 1 \right)$ and

$$
-\frac{\gamma}{2}\|\bar{\boldsymbol{\eta}}\|^2 \leq \begin{cases} -\left( \bar{F}(\boldsymbol{w}^t) - \bar{F}(\boldsymbol{w}^{ref}) \right)^2 / (2\gamma \|\boldsymbol{w}^{ref}\|^2) &, \text{if } \bar{F}(\boldsymbol{w}^t) - \bar{F}(\boldsymbol{w}^{ref}) < \gamma \|\boldsymbol{w}^{ref}\|^2 \\ -\frac{\gamma}{2}\|\boldsymbol{w}^{ref}\|^2 &, \text{o.w.} \end{cases} .
\tag{20}
$$

Note, since the function value $\{\bar{F}(\boldsymbol{w}^t)\}_{t=1}^{\infty}$ is non-increasing, only iterations in the beginning fall in second case of (20), and the number of such iterations is at most $c = \lceil \frac{2(\bar{F}(\boldsymbol{0}) - \bar{F}(\boldsymbol{w}^{ref}))}{\gamma \|\boldsymbol{w}^{ref}\|^2} \rceil$. For $t > c$, we have

$$
E[\bar{F}(\boldsymbol{w}^{t+1})] - \bar{F}(\boldsymbol{w}^t) \leq -\frac{\gamma \|\bar{\boldsymbol{\eta}}^t\|_2^2}{2} \leq -\frac{(\bar{F}(\boldsymbol{w}^t) - \bar{F}(\boldsymbol{w}^{ref}))^2}{2\gamma \|\boldsymbol{w}^{ref}\|^2}.
\tag{21}
$$

The recursion then leads to the result. □

Note the above bound does not yield useful result if $\|\boldsymbol{w}^{ref}\|^2 \to \infty$. Fortunately, the optimal solution of our target problem (7) has finite $\|\boldsymbol{w}^*\|^2$ as long as in (7) $\lambda > 0$, so it always give a useful bound when plugged into (18), as following corollary shows.

**Corollary 1** (Approximation Guarantee). *The output of Algorithm 1 satisfies*

$$
E\left[ \lambda \|\bar{\boldsymbol{w}}^{(D)}\|_1 + Loss(\bar{\boldsymbol{w}}^{(D)}; \boldsymbol{\phi}) \right] \leq \left\{ \lambda \|\boldsymbol{w}^*\|_2 + Loss(\boldsymbol{w}^*; \bar{\boldsymbol{\phi}}) \right\} + \frac{2\gamma \|\boldsymbol{w}^*\|_2^2}{D'}
\tag{22}
$$

*with $D' = \max\{D - c, 0\}$, where $\boldsymbol{w}^*$ is the optimal solution of problem (7), $c$ is a constant defined in Theorem 2.*

Then the following two corollaries extend the guarantee (22) to any $R \geq 1$, and a bound holds with high probability. The latter is a direct result of [18,Theorem 1] applied to the recursion (21).

**Corollary 2.** *The bound (22) holds for any $R \geq 1$ in Algorithm 1, where if there are $T$ iterations then $D = TR$.*

**Corollary 3.** *For $D \geq \frac{2\gamma \|\boldsymbol{w}^*\|^2}{\epsilon}(1 + \log \frac{1}{\rho}) + 2 - \frac{4}{c} + c$, the output of Algorithm 1 has*

$$
\lambda \|\bar{\boldsymbol{w}}^{(D)}\|_1 + Loss(\bar{\boldsymbol{w}}^{(D)}; \boldsymbol{\phi}) \leq \left\{ \lambda \|\boldsymbol{w}^*\|_2 + Loss(\boldsymbol{w}^*; \bar{\boldsymbol{\phi}}) \right\} + \epsilon
\tag{23}
$$

*with probability $1 - \rho$, where $c$ is as defined in Theorem 2 and $\boldsymbol{w}^*$ is the optimal solution of (7).*

## 3.2 Relation to the Kernel Method

Our result (23) states that, for $D$ large enough, the Sparse Random Features algorithm achieves either a comparable loss to that of the vanilla kernel method, or a model complexity (measured in $\ell_1$-norm) less than that of kernel method (measured in $\ell_2$-norm). Furthermore, since $\boldsymbol{w}^*$ is not the optimal solution of the $\ell_1$-regularized program (8), it is possible for the LHS of (23) to be much smaller than the RHS. On the other hand, since any $\boldsymbol{w}^*$ of finite $\ell_2$-norm can be the reference solution $\boldsymbol{w}^{ref}$, the $\lambda$ used in solving the $\ell_1$-regularized problem (8) can be different from the $\lambda$ used in the kernel method. The tightest bound is achieved by minimizing the RHS of (23), which is equivalent to minimizing (7) with some unknown $\tilde{\lambda}(\lambda)$ due to the difference of $\|\boldsymbol{w}\|_1$ and $\|\boldsymbol{w}\|_2^2$. In practice, we can follow a regularization path to find small enough $\lambda$ that yields comparable predictive performance while maintains model as compact as possible. Note, when using different sampling distribution $p(h)$ from the decomposition (3), our analysis provides different bounds (23) for the Randomized Coordinate Descent in Hilbert Space. This is in contrast to the analysis in the finite-dimensional case, where RCD with different sampling distribution converges to the same solution [18].

### 3.3 Relation to the Boosting Method

Boosting is a well-known approach to minimize infinite-dimensional problems with $\ell_1$-regularization [8, 9], and which in this setting, performs greedy coordinate descent on (8). For each iteration $t$, the algorithm finds the coordinate $h^{(t)}$ yielding steepest descent in the loss term

$$h^{(t)} = \underset{h \in H}{argmin} \quad \frac{1}{N} \sum_{n=1}^{N} L'_n \phi_h(\boldsymbol{x}_n) \tag{24}$$

to add into a working set $A^t$ and minimize (8) w.r.t. $A^t$. When the greedy step (24) can be solved exactly, Boosting has fast convergence to the optimal solution of (8) [13, 14]. On the contrary, randomized coordinate descent can only converge to a sub-optimal solution in finite time when there are infinite number of dimensions. However, in practice, only a very limited class of basis functions allow the greedy step in (24) to be performed exactly. For most basis functions (weak learners) such as perceptrons and decision trees, the greedy step (24) can only be solved approximately. In such cases, Boosting might have no convergence guarantee, while the randomized approach is still guaranteed to find a comparable solution to that of the kernel method. In our experiments, we found that the randomized coordinate descent performs considerably better than approximate Boosting with the perceptron basis functions (weak learners), where as adopted in the Boosting literature [19, 8], a convex surrogate loss is used to solve (24) approximately.

## 4 Experiments

In this section, we compare Sparse Random Features (Sparse-RF) to the existing Random Features algorithm (RF) and the kernel method (Kernel) on regression and classification problems with kernels set to Gaussian RBF, Laplacian RBF [2], and Perceptron kernel [7] [1]. For Gaussian and Laplacian RBF kernel, we use Fourier basis function with corresponding distribution $p(h)$ derived in [2]; for Perceptron kernel, we use perceptron basis function with distribution $p(h)$ being uniform over unit-sphere as shown in [7]. For regression, we solve kernel ridge regression (1) and RF regression (6) in closed-form as in [10] using Eigen, a standard C++ library of numerical linear algebra. For Sparse-RF, we solve the LASSO sub-problem (9) by standard RCD algorithm. In classification, we use LIBSVM[2] as solver of kernel method, and use Newton-CG method and Coordinate Descent method in LIBLINEAR [12] to solve the RF approximation (6) and Sparse-RF sub-problem (9) respectively. We set $\lambda_N = N\lambda = 1$ for the kernel and RF methods, and for Sparse-RF, we choose $\lambda_N \in \{1, 10, 100, 1000\}$ that gives RMSE (accuracy) closest to the RF method to compare sparsity and efficiency. The results are in Tables 1 and 2, where the cost of kernel method grows at least quadratically in the number of training samples. For *YearPred*, we use $D = 5000$ to maintain tractability of the *RF* method. Note for *Covtype* dataset, the $\ell_2$-norm $\|\boldsymbol{w}^*\|_2$ from kernel machine is significantly larger than that of others, so according to (22), a larger number of random features $D$ are required to obtain similar performance, as shown in Figure 1.

In Figure 1, we compare Sparse-RF (randomized coordinate descent) to Boosting (greedy coordinate descent) and the bound (23) obtained from SVM with Perceptron kernel and basis function (weak learner). The figure shows that Sparse-RF always converges to a solution comparable to that of the kernel method, while Boosting with approximate greedy steps (using convex surrogate loss) converges to a higher objective value, due to bias from the approximation.

#### Acknowledgement

S.-D.Lin acknowledges the support of Telecommunication Lab., Chunghwa Telecom Co., Ltd via TL-103-8201, AOARD via No. FA2386-13-1-4045, Ministry of Science and Technology, National Taiwan University and Intel Co. via MOST102-2911-I-002-001, NTU103R7501, 102-2923-E-002-007-MY2, 102-2221-E-002-170, 103-2221-E-002-104-MY2. P.R. acknowledges the support of ARO via W911NF-12-1-0390 and NSF via IIS-1149803, IIS-1320894, IIS-1447574, and DMS-1264033. This research was also supported by NSF grants CCF-1320746 and CCF-1117055.

Table 1: Results for Kernel Ridge Regression. Fields from top to bottom are model size (# of support vectors or # of random features or # of non-zero weights respectively), testing RMSE, training time, testing prediction time, and memory usage during training.

| | Gaussian RBF | | | Laplacian RBF | | | Perceptron Kernel | | |
|---|---|---|---|---|---|---|---|---|---|
| Data set | Kernel | RF | Sparse-RF | Kernel | RF | Sparse-RF | Kernel | RF | Sparse-RF |
| **CPU** | SV=6554 | D=10000 | **NZ=57** | SV=6554 | D=10000 | **NZ=289** | SV=6554 | D=10000 | **NZ=251** |
| $N_{tr}$ =6554 | RMSE=0.038 | 0.037 | **0.032** | 0.034 | . 0.035 | **0.027** | **0.026** | 0.038 | 0.027 |
| $N_t$ =819 | $T_{tr}$=154 s | 875 s | **22 s** | 157 s | 803 s | **43 s** | 151 s | 776 s | **27 s** |
| $d$ =21 | $T_t$=2.59 s | 6 s | **0.04 s** | 3.13 s | 6.99 s | **0.18 s** | 2.48 s | 6.37 s | **0.13 s** |
| | Mem=1.36 G | 4.71 G | **0.069** G | 1.35 G | 4.71 G | **0.095 G** | 1.36 G | 4.71 G | **0.090 G** |
| **Census** | SV=18186 | D=10000 | **NZ=1174** | SV=18186 | D=10000 | **NZ=5269** | SV=18186 | D=10000 | **NZ=976** |
| $N_{tr}$ =18186 | RMSE=**0.029** | 0.032 | 0.030 | **0.146** | 0.168 | 0.179 | **0.010** | 0.016 | 0.016 |
| $N_t$ =2273 | $T_{tr}$=2719 s | 1615 s | **229 s** | 3268 s | 1633 s | **225 s** | 2674 s | 1587 s | **185 s** |
| $d$ =119 | $T_t$=74 s | 80 s | **8.6 s** | 68 s | 88 s | **38s** | 67.45 s | 76 s | **6.7 s** |
| | Mem=10 G | 8.2 G | **0.55 G** | 10 G | 8.2 G | **1.7 G** | 10 G | 8.2 G | **0.49 G** |
| **YearPred** | SV=# | D=5000 | **NZ=1865** | SV=# | D=5000 | **NZ=3739** | SV=# | D=5000 | **NZ=896** |
| $N_{tr}$ =463715 | RMSE=# | **0.103** | 0.104 | # | 0.286 | **0.273** | # | 0.105 | 0.105 |
| $N_t$ =51630 | $T_{tr}$=# | 7697 s | **1618 s** | # | 9417 s | **1453 s** | # | 8636 s | **680 s** |
| $d$ =90 | $T_t$=# | 697 s | **97 s** | # | 715 s | **209 s** | # | 688 s | **51 s** |
| | Mem=# | 76.7G | **45.6G** | # | 76.6 G | **54.3 G** | # | 76.7 G | **38.1 G** |

Table 2: Results for Kernel Support Vector Machine. Fields from top to bottom are model size (# of support vectors or # of random features or # of non-zero weights respectively), testing accuracy, training time, testing prediction time, and memory usage during training.

| | Gaussian RBF | | | Laplacian RBF | | | Perceptron Kernel | | |
|---|---|---|---|---|---|---|---|---|---|
| Data set | Kernel | RF | Sparse-RF | Kernel | RF | Sparse-RF | Kernel | RF | Sparse-RF |
| **Cod-RNA** | SV=14762 | D=10000 | **NZ=180** | SV=13769 | D=10000 | **NZ=1195** | SV=15201 | D=10000 | **NZ=1148** |
| $N_{tr}$ =59535 | Acc=**0.966** | 0.964 | 0.964 | **0.971** | . 0.969 | 0.970 | **0.967** | 0.964 | 0.963 |
| $N_t$ =10000 | $T_{tr}$=**95 s** | 214 s | 180 s | **89 s** | 290 s | 137 s | **57.34 s** | 197 s | 131 s |
| $d$ =8 | $T_t$=15 s | 56 s | **0.61 s** | 15 s | 46 s | **6.41 s** | 7.01 s | 71.9 s | **3.81 s** |
| | Mem=3.8 G | 9.5 G | **0.66 G** | 3.6 G | 9.6 G | **1.8 G** | 3.6 G | 9.6 G | **1.4 G** |
| **IJCNN** | SV=16888 | D=10000 | **NZ=1392** | SV=16761 | D=10000 | **NZ=2508** | SV=26563 | D=10000 | **NZ=1530** |
| $N_{tr}$ =127591 | Acc=**0.991** | 0.989 | 0.989 | **0.995** | 0.992 | 0.992 | **0.991** | 0.987 | 0.988 |
| $N_t$ =14100 | $T_{tr}$=636 s | 601 s | **292 s** | 988 s | **379 s** | 566 s | 634 s | **381 s** | 490 s |
| $d$ =22 | $T_t$=34 s | 88 s | **11 s** | 34 s | 86 s | **25 s** | 16 s | 77 s | **11 s** |
| | Mem=12 G | 20 G | **7.5 G** | 12 G | 20 G | **9.9 G** | 11 G | 20 G | **7.8 G** |
| **Covtype** | SV=335606 | D=10000 | **NZ=3421** | SV=224373 | D=10000 | **NZ=3141** | SV=358174 | D=10000 | **NZ=1401** |
| $N_{tr}$ =464810 | Acc=**0.849** | 0.829 | 0.836 | **0.954** | 0.888 | 0.869 | **0.905** | 0.835 | 0.836 |
| $N_t$ =116202 | $T_{tr}$=74891 s | 9909 s | **6273 s** | 64172 s | 10170 s | **2788 s** | 79010 s | 6969 s | **1706 s** |
| $d$ =54 | $T_t$=3012 s | 735 s | **132 s** | 2004 s | 635 s | **175 s** | 1774 s | 664 s | **70 s** |
| | Mem=78.5 G | 74.7 G | **28.1 G** | 80.8 G | 74.6 G | **56.5 G** | 80.5 G | 74.7 G | **44.4 G** |

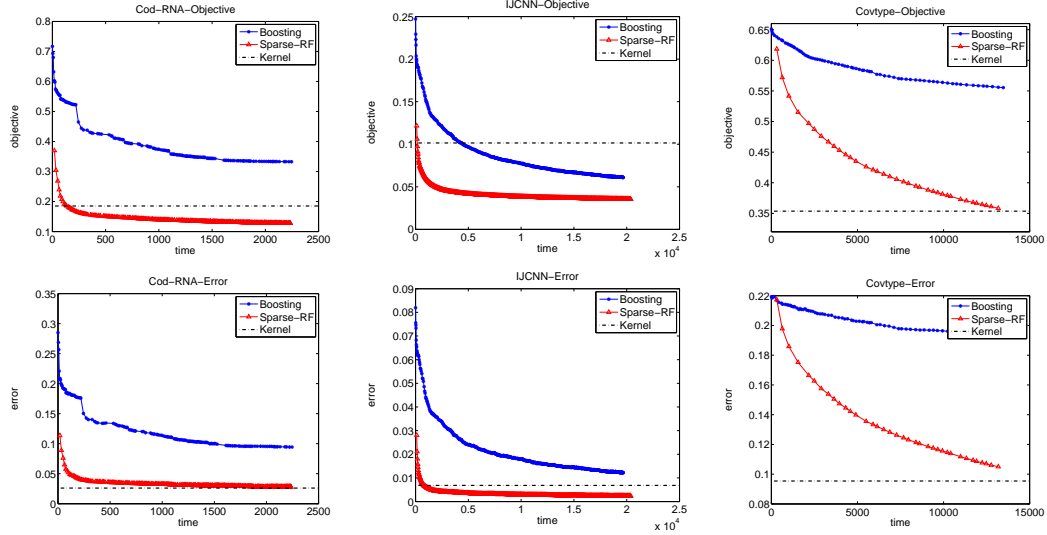

Figure 1: The $\ell_1$-regularized objective (8) (top) and error rate (bottom) achieved by *Sparse Random Feature* (randomized coordinate descent) and *Boosting* (greedy coordinate descent) using *perceptron* basis function (weak learner). The dashed line shows the $\ell_2$-norm plus loss achieved by kernel method (RHS of (22)) and the corresponding error rate using *perceptron kernel* [7].

## Footnotes

[2]Data set for classification can be downloaded from *LIBSVM data set* web page, and data set for regression can be found at UCI Machine Learning Repository and Ali Rahimi's page for the paper [2].

[2]We follow the FAQ page of LIBSVM to replace hinge-loss by square-hinge-loss for comparison.

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
