[Supplementary Material]

# Appendix: Sparse Random Feature Algorithm as Coordinate Descent in Hilbert Space

**Ian E.H. Yen** [1]     **Ting-Wei Lin** [2]     **Shou-De Lin** [2]     **Pradeep Ravikumar** [1]     **Inderjit Dhillon** [1]

Department of Computer Science

1: University of Texas at Austin,     2: National Taiwan University

1: {ianyen,pradeepr,inderjit}@cs.utexas.edu,

2: {b97083,sdlin}@csie.ntu.edu.tw

## 1  Proof of Lemma 1

**Lemma 1.** *Suppose loss function $L(z, y)$ has $\beta$-Lipchitz-continuous derivative and $|\phi_h(\boldsymbol{x})| \leq B, \forall h \in \mathcal{H}, \forall \boldsymbol{x} \in \mathcal{X}$. The loss term $Loss(\bar{\boldsymbol{w}}; \boldsymbol{\phi}) = \frac{1}{N}\sum_{n=1}^{N} L(\langle \bar{\boldsymbol{w}}, \boldsymbol{\phi}(\boldsymbol{x}_n)\rangle, y_n)$ in (9) has*

$$Loss(\bar{\boldsymbol{w}} + \eta\boldsymbol{\delta}_h; \boldsymbol{\phi}) - Loss(\bar{\boldsymbol{w}}; \boldsymbol{\phi}) \leq g_h\eta + \frac{\gamma}{2}\eta^2$$

*, where $\boldsymbol{\delta}_h = \boldsymbol{\delta}(\|x - h\|)$ is a Dirac function centered at $h$, $g_h = \nabla_{\bar{\boldsymbol{w}}} Loss(\bar{\boldsymbol{w}}; \boldsymbol{\phi})(h)$ is the Frechet derivative of loss term evaluated at $h$, and $\gamma = \beta B^2$.*

*Proof.*  For a loss function of $\beta$-Lipchitz-continuous derivative, we have

$$L(z + d, y) - L(z, y) \leq L'(z, y)d + \frac{\beta}{2}d^2 \tag{1}$$

. For $\bar{\boldsymbol{w}} + \eta\boldsymbol{\delta}_h$, we have $z + d = \langle \bar{\boldsymbol{w}}, \boldsymbol{\phi}(\boldsymbol{x}_n)\rangle + \eta\phi_h(\boldsymbol{x}_n)$. Substitute it into (1), average over $n$, apply the bound $|\phi_h(\boldsymbol{x}_n)| \leq B$, and the result follows.  □

## 2  Proof of Corollary 1

**Corollary 1** (Approximation Guarantee). *The output of Algorithm 1 has*

$$E\left[\lambda\|\bar{\boldsymbol{w}}^{(D)}\|_1 + Loss(\bar{\boldsymbol{w}}^{(D)}; \boldsymbol{\phi})\right] \leq \left\{\lambda\|\boldsymbol{w}^*\|_2 + Loss(\boldsymbol{w}^*; \bar{\boldsymbol{\phi}})\right\} + \frac{2\gamma\|\boldsymbol{w}^*\|_2^2}{D'} \tag{2}$$

*with $D' = \max\{D - c, 0\}$, where $\boldsymbol{w}^*$ is the optimal solution of problem (7), $c$ is a constant defined in Theorem 2.*

*Proof.*  Plug $\boldsymbol{w}^{ref} = \boldsymbol{w}^*$ into (18), we have

$$E[\bar{F}(\boldsymbol{w}^{(D)})] \leq \lambda\|\sqrt{\boldsymbol{p}} \circ \boldsymbol{w}^*\|_1 + Loss(\boldsymbol{w}^*; \bar{\boldsymbol{\phi}}) + \frac{2\gamma\|\boldsymbol{w}^*\|^2}{D'}, \tag{3}$$

where

$$\|\sqrt{\boldsymbol{p}} \circ \boldsymbol{w}^*\|_1 = \int_{h \in H} \sqrt{p(h)}|w^*(h)|dh \leq \sqrt{\int_{h \in H} p(h)dh}\sqrt{\int_{h \in H} w^*(h)^2 dh} = \|\boldsymbol{w}^*\|_2 \tag{4}$$

by Cauchy-Schwarz inequality and the fact probability distribution sums to 1.  □

# 3 Proof of Corollary 2

**Corollary 2.** *The bound (25) holds for any $R \geq 1$ in Algorithm 1, where if there are $T$ iterations then $D = TR$.*

*Proof.* We have proved the case when $R = 1$. To prove bound (25) for $R > 1$, we simply show that Algorithm 1 achieves larger descent amount if $R > 1$. Suppose current solution and working set are $\bar{w}^t$, $A^{(t)}$. Let $\bar{w}_1^{t+R}$, $A_1^{(t+R)}$ be solution and working set obtained from running Algorithm 1 for $R$ more iterations, each with 1 feature drawn, and let $\bar{w}_R^{t+1}$, $A_R^{(t+1)}$ be those obtained from running 1 iteration of Algorithm 1 with $R$ features drawn. From step 4 of Algorithm 1, we have $A_1^{(t+R)} \subseteq A_R^{(t+1)}$, and therefore $F(\bar{w}_R^{t+1}) \leq F(\bar{w}_1^{t+R})$ following step 3. $\qquad\square$