[Reviews · NeurIPS 2014]

Submitted by Assigned_Reviewer_2

The paper examines the problem of approximating Kernel functions by random features.
The main result is that using an L1 regularisation one can use only O(1/\epsilon) random features that to obtain an \epsilon accurate approximation to kernel functions.

The paper develops Sparse random features algorithm which is analogous to functional gradient descent in boosting.
The algorithm require O(1/\epsilon) random features which compares extremely favourably with the state of the art
which requires O1/\epsilon^2) features.

Detailed Convergence analysis are presented in the form of theorems. They appear to be correct.
The paper is well written.

This is an elegant result which should be of practical interest for solving large scale problems.
Summary: The proposed Sparse Random features algorithm is based on the idea of using L1 norm regularisation and yields significant improvement over existing work.

Submitted by Assigned_Reviewer_20

This paper is an incremental work on kernel approximation using Random Features. The proposed algorithm (Sparse Random Features) aims to introduce l1-regularization in the Random Feature algorithm so that the model doesn't grow linearly with the number of features. The authors also show that the proposed algorithm can be seen as a Randomized Coordinate Descent in Hilbert Space.

Overall the paper is easy to read and clear. The work appears to be significant and will allow for solving practical large scale problems more efficiently than current kernel methods.

However, I will suggest to make some efforts for a better presentation (analysis) of the experiments and results. There is a noticeable drop in accuracy of the proposed algorithm with Laplacian and Perceptron using the Covtype data, which could be further discussed. In addition to that, it is not clear whether the tables come from cross validation (preferable) or a simple split of the data.

Although the narrative is easy to follow and understand, I have the impression that some sentences need proofreading.
Summary: This is an interesting piece of work with direct impact on machine learning applications. The paper is well written, however, the authors require to improve the experiments section, and perhaps, adding conclusions and future work directions.

Submitted by Assigned_Reviewer_27

The paper presents a random feature based approximation algorithm for solving the l1 regularized problem in a (possibly infinite dimensinal) Hilbert space.

The paper is well written and easy to read. Theorem 2 and its corollaries are interesting and form the key technical contribution.

Solving l1 regularized problem in Hilbert space was considered ealier (for eg. [1*], which should perhaps be cited). However the proposed random feature algorithm and more importantly, its theoretical analysis are new and non-trivial.

Comments:
1. Prior work on solving l1 regularized problem in Hilbert spaces perhaps need to be summarized and cited. For e.g. [1*]. Also, alternatives like [2*] may be appropriately discussed.
2. The empirical section may be strengthened by comparing with [1*].
3. The current choices of \lamda seem arbitrary. Why not cross-validate for accuracy for all algorithms and report support-vectors/sparsity too? I think the current choice makes the values in the plots un-comparable. Or alternatively, provided plots with varying \lambda.
4. Some discussion on the results is needed, for e.g, performance in case of regression seems to be better than classification. why? etc.
[1*]. S. Rosset et.al. l1 regularization ininfinite dimensional feature spaces. COLT-2007.
[2*]. G. Song et.al. Reproducing kernel banach spaces with the l1 norm. Journal of Applied and Computational Harmonic Analysis.
Summary: The paper presents interesting theoretical results that establish the performance of a randomized feature based approximation algorithm for solving the l1 problem in a Hilbert space. However, the simulations section can be largely improved.
Author Feedback
Author rebuttal: We are thankful to all reviewers for their careful and constructive comments.

Reviewer_20:

Regarding the experiments section:

1. "There is a noticeable drop in accuracy of the proposed algorithm with Laplacian and Perceptron using the Covtype data, which could be further discussed."

The number of random features required for converging to similar performance of kernel method depends on \frac{\gamma\|w*\|^2}{\epsilon}. Here for Laplacian/Perceptron kernel \|w*\|^2 is actually larger than that of Gaussian kernel so one needs more random features to obtain similar performance. For Perceptron kernel, one can see that in Figure 1 (Covtype;Test-Error), Sparse-RF actually converges to similar performance to Kernel method after training for > 13000s, while in Table 2 we only show result of D=10000 features (1706s).

We will add clarifications to the final version as suggested.

2. "it is not clear whether the tables come from cross validation (preferable) or a simple split of the data."

They come from a simple random split of data, with train/test size specified as N_{tr}, N_{test} in the table. We will explicitly state this and add more explanations of experiment results in the final version.

Reviewer_27:

1. Regarding literature of infinite-dimensional L1-regularized problem [1*][2*].

The l1-regularized objective considered in [1*] is indeed the same as that considered here. The algorithm they proposed however is different, and requires a "greedy basis selection" step [1*,(7)], which is of the same spirit of, but more subtle than that of Boosting. Accordingly, how to approximately solve it would be an issue. Unlike the Spline basis considered in [1*], the basis functions considered here (Fourier,Perceptron) are more amenable to our approximate greedy steps.

The l1-regularized objective proposed in [2*], on the other hand, is different to ours. It directly penalizes the coefficients in the linear combination of kernel functions [2*,sec7]. We will add discussion of the references [1*][2*] along with Boosting in the final version.

[1*]. S. Rosset et.al. l1 regularization infinite dimensional feature spaces. COLT-2007.
[2*]. G. Song et.al. Reproducing kernel banach spaces with the l1 norm. Journal of Applied and Computational Harmonic Analysis.

2. Regarding the choice of \lambda in experiment.

Since Kernel and Random Feature (RF) optimizes the same objective function, their results are comparable by choosing the same \lambda=1. For Sparse-RF, which has a different objective, the current strategy is testing a variety of \lambda=100,10,1,0.1,0.01 to find the one with closest accuracy/RMSE to that of RF, and compare their computational efficiency.

Indeed, by choosing smaller \lambda (ex. 1) Sparse-RF can trade sparsity for better test accuracy/RMSE. We will provide plots of different \lambda to illustrate this trade-off in the final version.

3. "performance in case of regression seems to be better than classification. why?"

One reason is, for square-loss, the #Support Vector=#training samples, which makes exact kernel method more computationally demanding. On the other hand, there might also be the theoretical reason that the strict convexity of square-loss can lead to faster convergence of Sparse-RF algorithm, which should be further studied in the future work.